# Productivity of *Eucalyptus pellita* in Sumatra: *Acacia mangium* Legacy, Response to Phosphorus, and Site Variables for Guiding Management

**Eko B. Hardiyanto [1,*], Maydra A. Inail [2] and E. K. Sadanandan Nambiar [3]**

[1] Faculty of Forestry, Gadjah Mada University, Yogyakarta 55281, Indonesia
[2] PT. Musi Hutan Persada, Muara Enim, South Sumatra 31172, Indonesia; Maydra-Al@mhp.co.id
[3] CSIRO Land Water, GPO Box 1700, Canberra, ACT 2601, Australia; Sadu.Nambiar@csiro.au
[*] Correspondence: ebhardiyanto@ugm.ac.id

**Abstract:** We report on experimental studies conducted in South Sumatra with interrelated objectives to (i) examine the trends in production covering 30 years, including three rotations of *Acacia mangium* followed by *Eucalyptus pellita* which replaced *A. mangium* for managing the widespread threat of diseases; (ii) understand the effects of inter-rotation slash and litter management applied to acacia (legacy effects) on *E. pellita* growth; (iii) assess the long term changes in the top soil layer arising from above; (iv) evaluate, through a network of experiments, across the landscape, the nature and extent of growth responses to additional phosphorus. This data was also used to explore some of the critical site and stand variables which determine the variations in productivity and responses to management. The current growth rates of *E. pellita* are lower than those achieved in *A. mangium*. The management-legacy effects by conserving site resources provides a sustainable base for the growth of *E. pellita*, but for further increase in productivity, additional management actions are necessary. Changes in soil pH, carbon, N and extractable P were relatively small after four rotations. Supply of P at planting gave wood volume gains at harvest, ranging from 16 to 66% across sites. The plinthite layer in the soil profile was related to productivity, with higher growth rates of *E. pellita* occurring when the plinthite was at deeper layers. There is much scope for increasing productivity per unit area in this landscape, and available knowledge can be synthesized into a package of best practices for application. Management should aim to improve the quality of inter-rotation management to ensure more than 90% survival, and fast growth rates during the first 2 years. We provide a framework for further research and for refining management to produce the much needed additional domestic wood supply for the local industry.

**Keywords:** Indonesia; eucalypt and acacia plantations; production; multiple rotations; conservative-zero-till management; phosphorus response; soil-stand variables; adaptive management

## 1. Introduction

*Acacia mangium* was introduced in Sumatra, Indonesia, in 1979. Results from early studies showed that it had high growth rates in the local environment and soils, especially on *Imperata cylindrica* grass dominated, cut-over and degraded forest sites. Its fiber is highly suitable for pulp and paper. Large scale planting of the species commenced from about 1989, and by 2009 these plantations occupied about 565,000 ha in Sumatra and 379,000 ha in Kalimantan [E. B. Hardiyanto 2017, unpublished data from several sources] and became the primary source of wood for the pulp and paper mills in Indonesia.

In Sumatra, during the first rotation, *A. mangium* productivity in operational forestry, in 6–8 year rotations, typically ranged from 20 to 35 m³ ha⁻¹ y⁻¹, depending on site and management [1]. The growth rates in the second rotation were equal to or better than the first. In an experiment with a range of inter-rotation management treatments, the mean

growth rates in the second rotation at age 7 years reached 47.8 m$^3$ ha$^{-1}$ y$^{-1}$ compared to 29.7 m$^3$ ha$^{-1}$ y$^{-1}$ obtained at age 10 years in the first [2]. Such fast growth rates in response to improved management enabled the growers to harvest the same volume of timber from a reduced rotation length, from 7–9 years in the first rotation to 6–7 years in the second.

This trend in productivity was not sustained as the plantations progressed to third rotation. Since 1999–2000, root rot diseases caused by *Ganoderma philippii* fungus began to damage plantations [3,4] and became severe with successive rotations, causing 15–35% tree mortality, depending on site [5]. This was followed by a more devastating wilt disease caused by *Ceratocystis* spp. first reported by Tarigan et al. [6]. The spread of diseases was accelerated by stem wounds inflicted by groups of monkeys, squirrels and occasional entry of elephants; all of them peeled the bark creating entry points for the fungi. These diseases rendered *A. mangium* unviable in south and central Sumatra, and the industry responded by changing the species to *Eucalyptus pellita* at fast re-planting rates for meeting their wood supply [7]. Based on the current experience, *E. pellita* has a higher resistance to these diseases and has the potential to grow well in Sumatra. Currently, *E. pellita* had replaced *A. mangium* in about 465,000 ha in Sumatra and 225,500 ha, in Kalimantan. This large change has occurred in less than 10 years in plantations owned by the companies and also by the many medium and small growers who sell their wood to one of the six pulp mills in Sumatra [8].

The growth rates of *E. pellita* in relatively small areas planted earlier, were in the range 15.6–17.6 m$^3$ ha$^{-1}$ y$^{-1}$, significantly lower than that of *A. mangium* [1]. It should be noted that these eucalypt plantations were established when local managers had limited experience with eucalypts. Recent inventory data of MHP Company plantations showed improvements, but still recorded a 24–36% lower volume productivity, at a broad scale, compared to *A. mangium* at the same sites [Unpublished data, 2021, MHP, South Sumatra]. The change in species and the current trends in productivity have major impacts on wood supply for local processing and in turn on regional and national economy. The current pulp production capacity of the six mills in Sumatra is 10.25 million tonnes y$^{-1}$ which requires close to 42 M m$^3$ y$^{-1}$ wood at the mill [9]. Nambiar et al. [7] estimated the production of 10.1 million tonnes of air-dry pulp requires harvesting a standing volume of about 51 M m$^3$ y$^{-1}$ of *A. mangium* accounting for the unavoidable losses from standing trees to debarked logs at the mill; the equivalent standing volume from *A. crassicarpa* or from *E. pellita* and its hybrids would be somewhat different depending on bark percentage, wood density and pulp yield. In 2019, harvested log volume in Sumatra was 35.54 M m$^3$ consisting of *A. crassicarpa* (28.57 M m$^3$) and *E. pellita* (6.97 M m$^3$). Consequently, there is a significant wood deficit, the amounts depending on the estimates. To meet this shortage the pulp mills are transporting wood over long distances from Kalimantan where estimated total production was 6.04 M m$^3$ [10] and importing from other counties.

All pulp and paper companies in Sumatra are committed to procuring wood exclusively from plantations, and none are planning to expand their production area because of social and political challenges. Furthermore, some have reallocated significant production areas for conservation and other environmental values. Thus, it is an imperative that productivity per unit area is increased sustainably to ensure the stability of wood supply and to manage the fluctuations in production that occur in response to vagaries of climate and biological stresses.

Nambiar et al. [7] identified the critical issues that would help to increase the wood productivity by practicing an integrated approach to applications of R&D outcomes from Indonesia and comparable environment [11,12]. These include conservation of site/soil resources [2,13], breeding and deployment of the best germplasm [14], vegetation and nutrient management [15,16] and improved working and reward conditions for the workforce [7].

In this paper, we present the results based on stand growth to full commercial rotation length from a network of seven experiments located at sites representative of the landscapes in South Sumatra. The main objectives were to

- Examine the long term trend in productivity over 30 years, and across four successive rotations (three rotations of *A. mangium*, followed by *E. pellita*) at the same site.
- Evaluate the impacts of successive applications of management practices representing depletion or retention of slash and litter over the three previous rotations of *A. mangium*, (legacy-effects) on productivity of *E. pellita.*
- Examine the question, what is the nature of growth responses to P (rates of application and the extent across sites) to *E. pellita* planted on ex. *Acacia* sites, given the history of retention of slash and litter, and the application of P to previous *A. mangium* rotations?
- Using the data from the network of experiments across a range of sites and contrasting growth rates, examine how the site and stand influence the variation in site productivity.

We also note that due to a variety of constraints, few, if any, published studies e.g., [15–17] on *E. pellita* have followed growth of stands to a full rotation and harvest. Here, we present the first set of results on the impacts of management and sites on productivity over the full *E. pellita* rotation in Indonesia. The results are then discussed in relation to potential application of results, at operational scale, within the company estate and by the many small growers operating in the region.

## 2. Materials and Methods

### 2.1. Productivity over Successive Rotations

This site is a focus of a long term study at Sodong, South Sumatra. It builds on the first phase of a study on *A. mangium* [2,18] and now examines the changes in productivity from *A.mangium* to *E. pellita* and soils. The site is located at 10°00′01″ E and 3°41′02″ S at an altitude of 80 m above sea level. It has flat topography and consists of Ultisol soils [19] derived from sedimentary rock consisting of sandy tuff, sandstone and clay stone. The site was previously dominated by *Imperata* grass before clearing and establishment of acacia plantations. The site history and properties of soil have been reported previously [18].

#### 2.1.1. Treatments

After harvesting the first rotation (R-1) stand of *A. mangium,* the following treatments were applied for the second rotation (R-2) of *A. mangium* [2]:

$BL_0$　All above ground biomass removed: All merchantable stems (more than 7 cm top-end diameter, over bark) slash, litter and understorey were removed.
$BL_1$　Whole tree harvested: All trees with merchantable wood including crown were removed from the plots. Litter and understorey were retained.
$BL_2$　Merchantable stems harvested: Remaining stem top-ends, branches, leaves, litter and understorey were retained and re-distributed evenly in the same plot.
$BL_3$　High organic matter: Same as $BL_2$ but slash from the $BL_1$ plots was brought in and distributed evenly over the slash already present in $BL_2$ plots. This created a high organic matter treatment for comparison. This practice would not occur in operation, but a harvested site would have many small piles of slash giving rise to patches of high organic matter

The experiment was laid out as a randomized complete block design, with four replications. Plot size was 36 × 36 m, providing 100 measured trees and one border row. These treatments were repeated in each plot at every rotation from the establishment of R-2 to R-4, manually to minimize soil disturbance.

#### 2.1.2. Management

First rotation (R-1): *A. mangium* was planted in January 1991, harvested at age 10 years. The site was ploughed and harrowed, as per the common practice at that time. Seedlings raised from unimproved seed were planted at spacing of 3 × 2 m (1666 trees

ha$^{-1}$) in pits. A dose of fertilizer mixture (4.6 g N, 2.0 g P and 5.0 g K tree$^{-1}$) was applied one month after planting, around the base of the trees.

Second rotation (R-2): Re-planted with *A. mangium* in January 2001 and harvested at age 7 years. Seedlings raised from seeds obtained from a local seed orchard were planted at a spacing of 3 × 3 m (1111 trees ha$^{-1}$). Triplesuperphosphate (14 g P tree$^{-1}$) was placed in the planting hole, and Urea (13.8 g N tree$^{-1}$) was applied one month after planting, 10 cm away from the base of the seedling.

Third rotation (R-3): Planted with *A. mangium* in January 2009. In addition to the four core treatments four additional plots (called BL$_{2+P}$) were established. The reasons for including this treatment were evidence of response to P in *A. mangium* during early growth at some sites [20] and a steady decline in extractable soil P with time in acacia in Sumatra [2] and South Vietnam [21]. For this treatment, we used the plots earlier set up for biomass sampling from each replication. They then received triple-superphosphate (40 kg P ha$^{-1}$) at planting, put at the bottom of the planting hole. The stand was damaged severely by *Ceratocystis* wilt disease and monkeys. Therefore, the last growth measurements were taken at age 3 years. The stand was harvested at 4.5 years and some logs were salvaged leaving all slash in plots.

Fourth rotation (R-4): Planted with *E. pellita* in January 2014 and harvested at 6 years of age. Treatments were the same as in R-3. Seedlings were raised from seed orchard seeds and were planted at a spacing of 3 × 2 m (1666 trees ha$^{-1}$).

*2.2. Responses to Phosphorus and Site Variation in E. pellita*

There were six sites within a geographical area spanning from 03°37′58.6″ S, 103°36′58.5″ E to 03°48′30.5″ S, 103°56′29.0″ E (Table 1). Soils at all sites belong to Ultisols. The soils varied in their depth of A-horizon and to the sub-soil plinthite zone; the latter property influences the growth of *A. mangium* [22]. The climate of the region is humid with the average daily temperature of about 29 °C and the average minimum and maximum temperatures of 22 and 32 °C, respectively. The annual rainfall during the last 20 years ranged from 1816 to 3885 mm, falling mainly between October and May with a relatively dry period from June to September (in some years with rainless months) when trees generally experience water stress, especially on shallow soils. All sites had 2–3 rotations of *A. mangium* previously and were located across the commercial plantation forestry region. After harvesting the salvageable timber (all were affected by diseases and low volumes of timber were harvested at age 3–4 years), slash and litter were left on site.

**Table 1.** Soil profile features and properties of 0–10 cm layer, at sites where responses to P were tested (see Figure 5).

| Properties | Site | | | | | |
|---|---|---|---|---|---|---|
| | Merbau-3 | Nakau | Merbau-1 | Lengi | Lantingan | Niru |
| Slope (%) | 3–8 | 3–6 | 30–40 | 3–5 | 5–8 | 8–10 |
| Bulk density (g cm$^{-3}$) | 1.07 | 1.02 | 1.16 | 0.90 | 1.40 | 1.17 |
| Depth of A-horizon (cm) | 12 | 15 | 14 | 9 | 10 | 3 |
| Depth to plinthite layer (cm) | 109.9 | 83.7 | 79.9 | 70.4 | 66.9 | 34.2 |
| Sand (%) | 56.7 | 44.6 | 46.7 | 31.9 | 40.8 | 37.4 |
| Silt (%) | 22.1 | 31.3 | 27.5 | 32.1 | 40.6 | 36.3 |
| Clay (%) | 21.2 | 24.1 | 25.8 | 35.9 | 18.6 | 16.3 |
| pH (H2O) | 4.3 | 4.2 | 4.2 | 4.3 | 4.1 | 4.2 |
| Total N (%) | 0.12 | 0.15 | 0.12 | 0.23 | 0.14 | 0.18 |
| Extractable P (mg kg$^{-1}$) | 2.76 | 2.26 | 3.63 | 2.19 | 3.09 | 2.16 |

Experiments were planted with three-month old seedlings of *E. pellita* raised from seeds collected from a local seed orchard. All sites were planted in November–December 2013. The experiments were laid out as a randomized complete block design with four replicates. There were 49 trees per plot, providing 25 measured trees and one border row. The spacing was 3.0 × 2.5 m (1333 trees ha$^{-1}$). The treatments aimed to create a wide range

in P supply. Triplesuperphosphate was applied at rates of 0, 12, 24, 36 and 48 kg P ha$^{-1}$; fertilizer was placed at the bottom of the planting holes.

All experiments were managed under a common set of practices: conservation of site organic matter, seedlings from one source, common stocking, and judicious weed control. Weeds were managed by application of 2.5 L ha$^{-1}$ of glyphosate at age 3 months, glyphosate + metsulfuron-methyl (150 g ha$^{-1}$) at age 6 months. After this, herbicides were applied six-monthly until age 2 years, and then annually until the final year of the rotation.

### 2.3. Growth Measurements

Height and diameter of trees were measured, annually but here we report only the results at the end of rotation. Volumes were estimated using the equations:

*A. mangium*: Total stem volume (*v*) = 0.25 $\pi$ $D^2$ $H$ 0.45

*E. pellita*: Total stem volume (*v*) = 0.25 $\pi$ $D^2$ $H$ 0.48

where *D* is diameter at 1.3 m height in cm, *H* is total height in m. Form factors were based on previous studies in the region; 0.45 for *A. mangium* [23] and 0.48 for *E. pellita* [24]. Total volume per ha (*V*) for each treatment was calculated as the sum of individual tree volume (*v*) per plot converted to a per hectare basis.

### 2.4. Soil Sampling and Analysis

Soil in 0–10 cm layer was sampled (10 cores per plot and bulked) in the successive rotation experiment, at the end of each rotation from R-1 to R-4. Samples were air dried, sieved, and the fractions less than 2 mm were subsampled for analysis. Soil organic carbon (SOC) was determined by $K_2Cr_7$ and $H_2SO_4$ digestion and measured by spectrophotometer; total N, by $H_2SO_4$ digestion and Kjeldhal; extractable P (Bray-I) by colorimetry, exchangeable K, Ca, and Mg in $NH_4Ac$ extracts at pH 7.0 in 1:2.5 water suspension. In the P-response experiments, samples were taken before planting. The sampling and preparation described earlier were followed.

At all sites, the depth to plinthite/gleyed horizon and depth of A-horizon were measured using soil borer, at 8 random points within each replication.

### 2.5. Statistical Analyses

A two-way analysis of variance was performed to determine significance of difference between treatments, followed by Duncan multiple range test at $\alpha$ = 0.05 if differences between treatments were significant. All analyses including non-linear regression to fit P rate response were performed using R.ver. 4.02 software.

## 3. Results

### 3.1. Productivity over Four Rotations and the Comparative Growth of Species

There was a substantial increase in productivity of *A. mangium* from R-1 to R-2 (Figure 1). This is attributable to improvements in management, including the planting of genetically selected seedlings, conservation of organic matter and nutrients, application of P and weed management. Productivity in R-2 (*A. mangium*) was markedly higher than in R-4 (*E. pellita*) under similar slash and litter management regimes, despite R-2 having lower initial stocking (1111 trees ha$^{-1}$) than R-4 (1666 trees ha$^{-1}$).

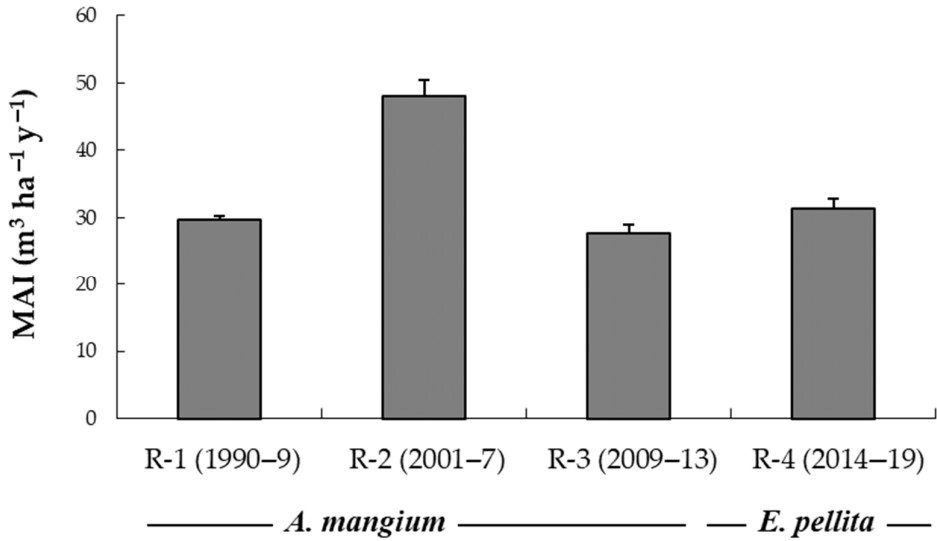

**Figure 1.** Wood production measured as mean annual increments (MAI) in four successive rotations. *Acacia mangium* in the first three and *Eucalyptus pellita* in the fourth. Vertical bars are SE of means.

The third rotation of *A. mangium*, although established under the same management regime as in R-2, had to be terminated as an experiment at 3 years of age, due to severe tree losses caused by *Ceratocystis* fungi. It was harvested at age 4.5 years and the productivity was very low (Figure 1). At a common age of 6 years, the growth rate of *E. pellita* was substantially lower (by 43.3% in stem volume) than that of *A. mangium* in R-2. All growth parameters reflected this pattern (Figure 1; Table 2).

**Table 2.** Growth parameters of *Acacia mangium* in the second rotation and *Eucalyptus pellita* in the fourth rotation, both at age six 6 years.

| Parameter | *Acacia mangium* | *Eucalyptus pellita* |
|---|---|---|
| Stocking: planting/harvest (tree ha$^{-1}$) | 1111/869 | 1666/1005 |
| Height (m) | 26.4 (0.2) | 18.4 (0.3) |
| Diameter (Dbh, cm) | 20.3 (0.9) | 13.0 (0.1) |
| MAI (m$^3$ ha$^{-1}$ y$^{-1}$) | 50.0 (1.5) | 34.9 (0.7) |
| Commercial wood, diameter > 5 cm (m$^3$ ha$^{-1}$) | 288.3 (8.8) | 178.3 (3.3) |
| Standing biomass (Mg ha$^{-1}$) | 171.6 (4.9) | 131.9 (3.0) |
| Mean litterfall (Mg ha$^{-1}$y$^{-1}$) | 10.0 (0.4) | 8.7 (0.4) |
| Rain during the rotation (mm y$^{-1}$) | 2570 | 2848 |

Numbers in the parenthesis are SE. Data of *A. mangium* is from BL$_2$ treatment and *E. pellita* from BL$_{2\ +P}$ treatment.

The effect of conserving organic matter (slash and litter) in the previous three successive rotations of *A. mangium* (legacy-effect) on *E. pellita* was modest. The retention of slash and litter significantly increased tree growth (height, $p = 0.036$; diameter, $p = 0.001$ and stem volume, $p = 0.027$) up to age 3 years (data not given here), thereafter its effect was statistically not significant, except for the difference between BL$_0$ (repeated removal) and BL$_3$ (double slash + litter). At age 3 years, the mean height for BL$_0$ treatment was $11.9 \pm 0.1$ m, diameter $9.4 \pm 0.1$ cm, stem volume $68.6 \pm 1.2$ m$^3$ ha$^{-1}$, while for BL$_3$ treatment the corresponding values were $12.6 \pm 0.3$ m, $10.2 \pm 0.2$ cm and $81.8 \pm 2.4$ m$^3$ ha$^{-1}$, and at age 6 years the volumes for BL$_0$ and BL$_3$ were $176.4 \pm 4.2$ and $187.9 \pm 2.7$ m$^3$ ha$^{-1}$, respectively.

The additional growth response to added P (BL$_{2+P}$) by *E. pellita* was strong despite the history of organic matter retention and P applications in the previous rotations. The additional P gave 8% (not significant) more stem volume than BL$_2$, and 19% ($p < 0.05$) more compared to the treatment where slash and litter were removed (BL$_0$) (Figure 2). At harvest, BL$_3$ and BL$_{2+P}$ had lower stocking with a mean of 1232 trees ha$^{-1}$ compared with other treatments which were similar at 1300 trees ha$^{-1}$, but this effect was not statistically significant.

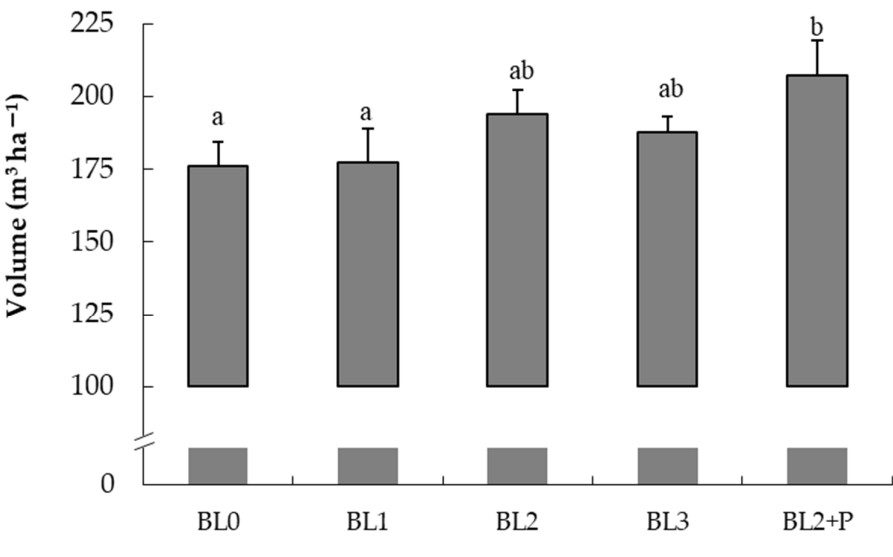

**Figure 2.** The legacy effects of inter-rotation management over three successive rotations of *Acacia mangium* on stem volume of *Eucalyptus pellita* at age 6 years. Vertical bars are SE of means. Treatments having the same letter are not significantly different according to Duncan multiple range test at $\alpha = 0.05$.

The stem volume growth of R-2 (*A. mangium*) and R-4 (*E. pellita*) are compared in Figure 3. For simplifying the presentation, we used the mean of BL$_0$ and BL$_1$ (low organic matter) and the mean of BL$_2$ and BL$_3$ (high organic matter). In both species, the stem volume began to diverge at about age 2 years, indicating that organic matter retention promoted faster growth in *A. mangium* than in *E. pellita*. The growth differences between species were maintained throughout the rotation.



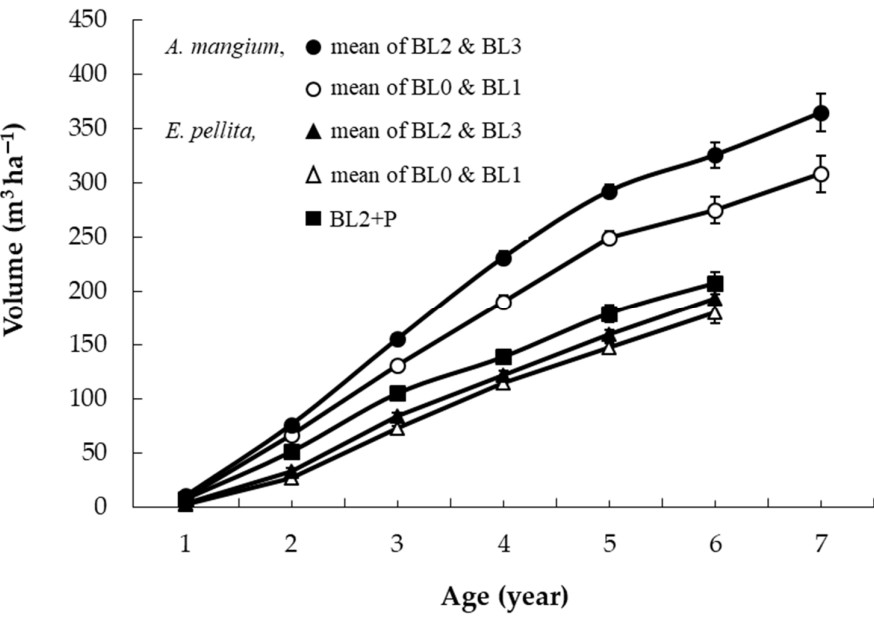

**Figure 3.** Growth response of *Acacia mangium* and *Eucalyptus pellita* at the same site, to inter-rotation management. Acacia data is for the second rotation (from Hardiyanto and Nambiar [2]) and eucalypt data is from the fourth rotation. Vertical bars are SE of means.

### 3.2. Changes in Soil Properties across Successive Rotations

The changes in soil properties (0–10 cm) from the end of every rotation from R-1 (*A. mangium*) to R-4 (*E. pellita*) are shown in Figure 4. Properties of treatments $BL_0$ and $BL_3$ are shown as they represent the two extreme slash and litter management treatments.

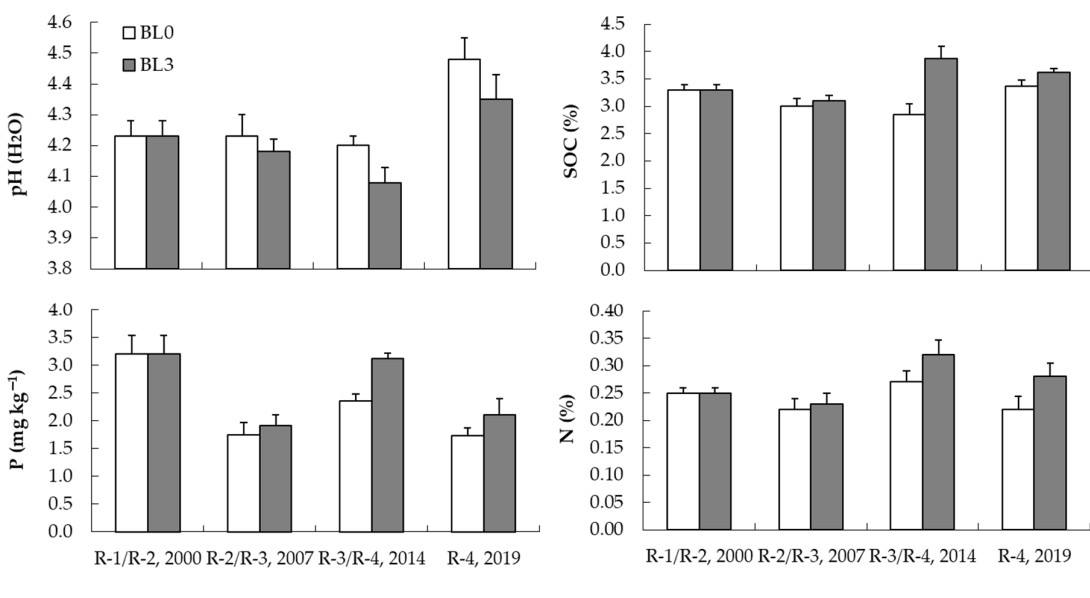

**Figure 4.** Changes in 0–10 cm soil layer: pH, organic carbon (SOC), extractable P, and total N in from the end of the first rotation (R-1/R-2) to the end of fourth rotation (R-4). Vertical bars are SEs. R-1 to R-3 *Acacia mangium*, and R-4 *Eucalyptus pellita*.

Soil pH at the end of R-4 was higher both in $BL_0$ and $BL_3$ treatments compared with that of R-1; increases were about 0.25 and 0.12 units in $BL_0$ and $BL_3$, respectively. Soil organic carbon (SOC) values were always higher in $BL_3$ (double slash and litter retention) than $BL_0$ across successive four rotations. In both cases, SOC concentrations fluctuated to which sampling and analytical errors may have contributed. At the end of R-4 the SOC was about 10.0% higher than in R-1. Soil N showed a trend similar to SOC, however, at the end of R-4 the N level in $BL_3$ was slightly higher (12.0%) than at the end of R-1, while N level in $BL_0$ showed little change. In $BL_0$ and $BL_3$ extractable soil P declined steadily towards the end of R-2, increased slightly in R-3, and then declined at the end of R-4. Extractable P concentration was higher in $BL_3$ than in $BL_0$ across four successive rotations.

### 3.3. Responses to Phosphorus by E. pellita

There were positive responses to applied P at all sites from early stages of stand growth. For example, at all sites, at age 2 years, P application resulted in significant positive responses in tree height ($p = 0.011$ - <0.001), stem diameter ($p = 0.002$ - <0.001), stem volume ($p = 0.003$ - < 0.001) (data not shown), and this trend continued in volume growth at age 6 years, the common harvesting age in Sumatra.

It should also be noted that at 6 years of age, the interactions between site and dose of P were significant for height ($p = 0.001$), diameter ($p = 0.039$) and stem volume ($p = 0.007$) (details not given). Furthermore, the relative size of the response to P declined with age at all sites. For example, at 2 years of age, at 12 kg P ha$^{-1}$, increases in stem volumes ranged from 42% to 318% over control (no added P), while at age 6 years the corresponding values ranged from 12% to 44%; equivalent to volume growth from 17.3 to 64.4 m$^3$ ha$^{-1}$. The declining response to P with age was partly dependent on the level of site productivity, indicating that the response to P depends on site quality.

Figure 5 shows the regression fits for P-rate responses. At all sites, tree growth improved significantly when P dose was increased from 0 to 12 kg P ha$^{-1}$, and the further increases in P provided additional growth at three sites: Merbau-1, Merbau-3 and Lantingan, but not at Nakau, Lengi and Niru. At a dose of 12 kg P ha$^{-1}$ mean annual increment (MAI) ranged from 25.3 to 31.9 m$^3$ ha$^{-1}$ y$^{-1}$ at age 6 years between sites. The patterns of response curves in Figure 5 suggest a potential grouping of sites based on productivity. For simplicity, we consider Merbau-1 and Merbau-3 as high productivity, Lantingan and Nakau as medium productivity and Lengi and Niru as low productivity.

The estimated 95% of the maximum volume yield for the high, medium and low productivity groups, as means of pairs of sites in each were 207, 165 and 147 m$^3$ ha$^{-1}$, respectively, and the corresponding P rates of applications were, 24.7, 13.2 and 7.3 kg P ha$^{-1}$. At the high productivity sites, response to P continued beyond the 12 kg ha$^{-1}$ rate, while at the two low productivity sites there was no additional growth from higher doses (Figure 5). The medium productivity sites were somewhat in the middle. At the high productivity sites, increasing P dose from 12 to 24 kg ha$^{-1}$ yielded additional volumes of 14.1 and 22.4 m$^3$ ha$^{-1}$, whereas at the low productivity sites, there were no improvements.

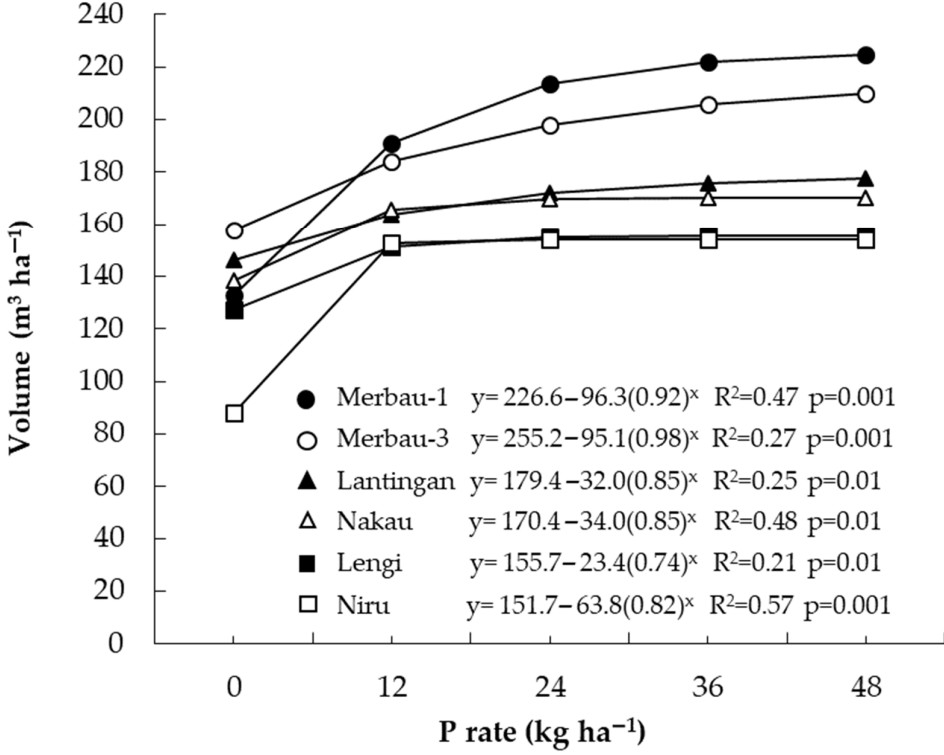

**Figure 5.** Productivity of *Eucalyptus pellita* at age 6 years: The responses in stem wood production to phosphorus applied at planting.

### 3.4. Site Variations in Productivity

The productivity of *E. pellita* varied substantially between sites (Figure 5) with or without P addition; the highest yield with P application was 224.8 m³ ha⁻¹ at Merbau-1 compared to 154.0 m³ ha⁻¹ in Niru. What are the factors influencing this, given that all sites were planted with the same planting stock within 4 weeks, and the site and stands were managed in a similar manner? The rates of mortality varied between sites and, therefore, the survival at the end of the rotation (Figure 6). Mortality increased with age at all sites; at the end of 6 years, stocking ranged from 934 to 1096 trees ha⁻¹, markedly lower than at planting 1333 trees ha⁻¹. At age 6 years, the effect of added P on survival rates varied, dependent on sites and was statistically significant ($p = 0.050$). At three sites, plots supplied with P had significantly ($p = 0.006$) higher survival (77.3 ± 1.7%) compared to those under P-0 (67.4 ± 2.1%). At two sites P had no effect, 72.5 ± 2.6% and 72.3 ± 2.8 % with and without P, respectively. At one site survival was reduced with added P. There is an indication that application of P could improve survival on some sites. However, growth response to P (Figures 5 and 6) can be largely attributed to improved growth rates (current annual increments, CAIs). For example, without added P, the average CAI across sites from age 5 to 6 years was 27.1 m³ ha⁻¹ y⁻¹, whereas with P addition the corresponding CAI was 38.5 m³ ha⁻¹ y⁻¹.

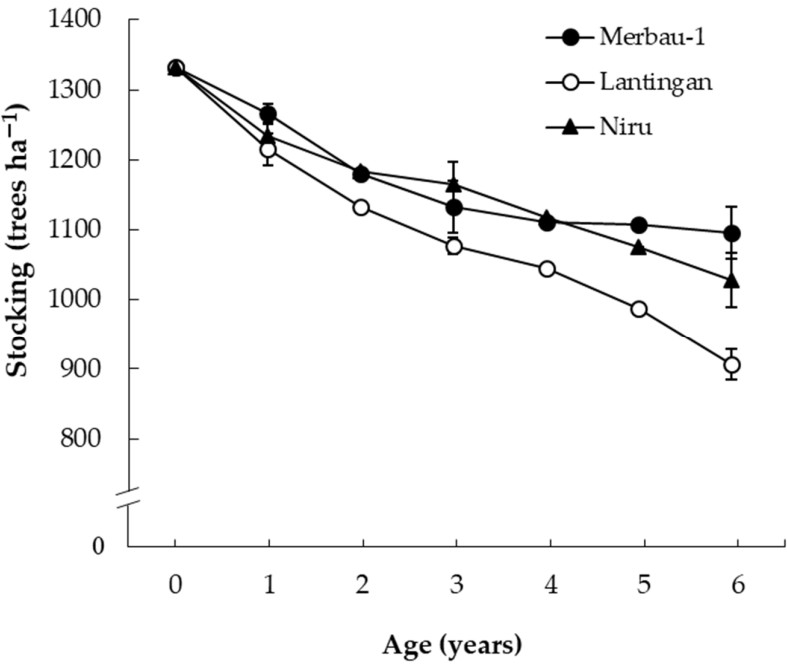

**Figure 6.** Changes in stocking (trees ha⁻¹) of *Eucalyptus pellita* from planting to harvest at sites representing three levels of productivity: high (Merbau-1), medium (Lantingan) and low (Niru). Vertical bars are SE of means of four replications. Data was from pooled-data of added P treatment.

Stocking rate had a large effect on final wood production (Figure 7). The nature of the effects of P on growth is further evident in Figure 7a,b. When P supply was adequate for growth, there was a strong positive linear relationship ($R^2 = 0.58$, $p < 0.001$) between stocking and stem volume at harvest, while under low P supply (controls), there was a trend but the relationship was not significant ($R^2 = 0.03$, $p = 0.47$). In general, our results illustrate the critical importance of ensuring survival and this should be a priority for management attention.

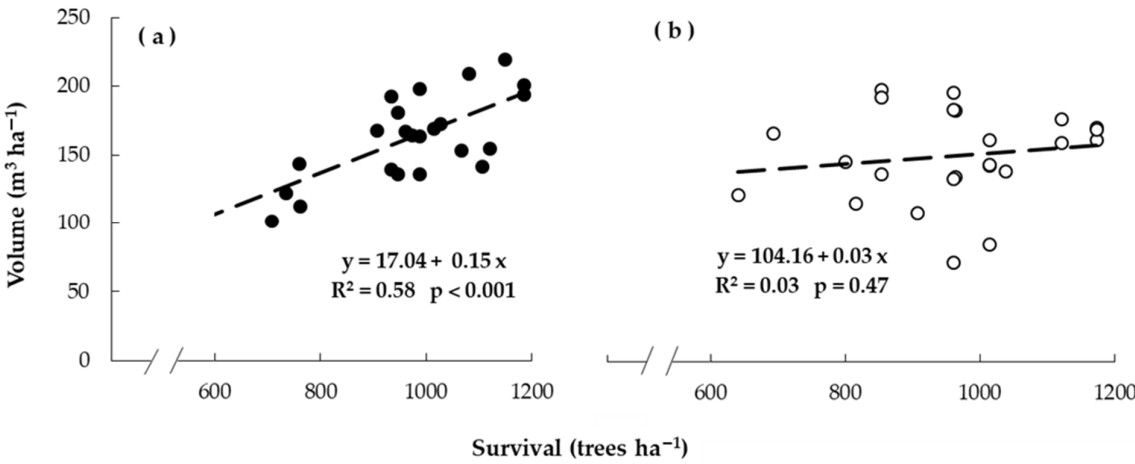

**Figure 7.** Volume of *Eucalyptus pellita* at age 6 years: relationship between tree survival and stem volume: (**a**) with P fertilizer (pooled-data of added P treatments), (**b**) without P fertilizer. Data points represent individual plots from all replications and respective treatments from six sites.

We explored the effects of A-horizon and the depth to impeding horizon (plinthite/gleyed) in the profile on growth. In Figure 8a,b we used volume data from P-0 plots because the relative response to P varied between sites (Figure 5). Both, depth to plinthite and depth of A-horizon influenced growth (Figure 8a,b), the deeper the plinthite horizon the higher the stem volume ($R^2 = 0.61$, $p < 0.001$). Similarly, the thicker the depth of A-horizon more favorable it is for growth. The relationship was much stronger with the depth to plinthite than with the depth of A-horizon. Further analyses revealed (data not shown) that survival was not influenced by the depth of A-horizon or the depth to plinthite horizon. These suggest that the soil depth influences productivity largely by the effects on tree growth rates.

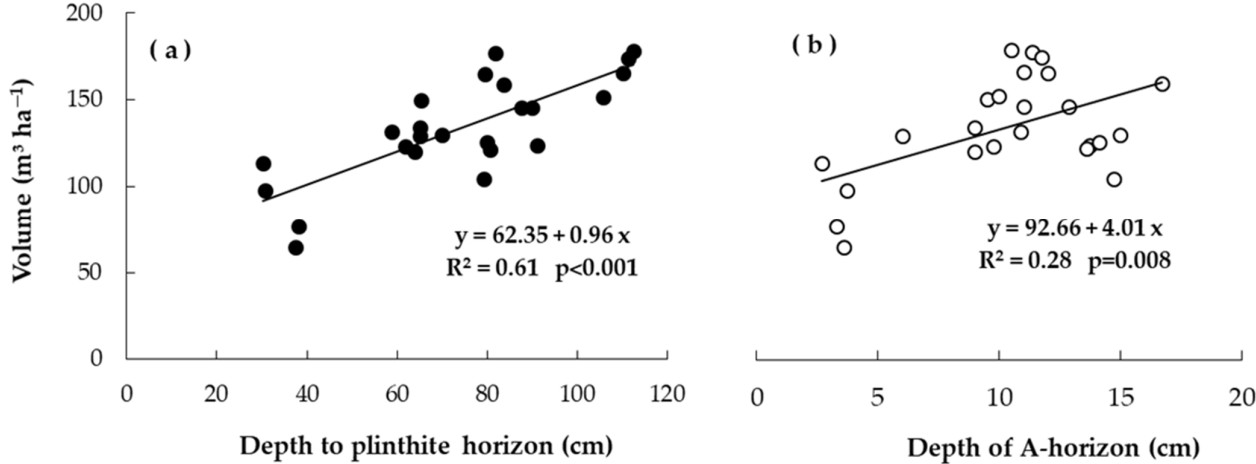

**Figure 8.** *Eucalyptus pellita* at age 6 years: relationship between stem volume and (**a**) depth to plinthite horizon, (**b**) depth of A-horizon. Data points represent individual plots of P-0 treatment from all replications and six sites.

*3.5. Relationship between Early Growth and Yield at Harvest*

In short rotation forestry, the early growth, survival and the establishment of the stand (site capture and ability to compete with weeds) is known to be a key determinant of yield [12,25]. This was explored for *E. pellita* under the local conditions (Figure 9). For this, we used basal area as it integrates stocking and stem diameter growth data at 12 kg P ha$^{-1}$ y$^{-1}$ (Figure 5). Figure 9 shows a strong relationship between basal area at age 2 years and volumes at age 6 years. When P supply was low (no added P) the relationship followed a non-linear trend ($R^2 = 0.52$, $p = 0.01$), indicating a drop in current annual increment (CAI) as the stand grew. With higher P supply, the relationship was linear ($R^2 = 0.62$, $p < 0.001$) indicating sustained increase in CAI. For example, without added P the average CAIs between two growth periods, ages 4–5 and ages 5–6 years were 27.6 and 27.1 m$^3$ ha$^{-1}$ y$^{-1}$, respectively; in comparison, with added P the corresponding CAIs were 34.1 and 38.5 m$^3$ ha$^{-1}$ y$^{-1}$, respectively. The relationships shown in Figures 9a,b were similar with volumes at age 2 (data not shown), but basal area is preferable because it does not require height measurements which are more time consuming and error prone than stem diameter. After age 2 years, the rate of mortality also declines (Figure 6) and the stand is beyond the establishment phase.

There is another important observation in Figure 9: in the P deficient plots (P-0) basal area ranged from 0.95 to 6.64 m$^2$ ha$^{-1}$, a nearly 7-fold difference (mean 3.99 ± 0.30 m$^2$ ha$^{-1}$, CV 37.2%, $n = 24$), indicating high variation and uneven stand growth. In contrast, with adequate P (12 kg P ha$^{-1}$) supply, the basal area ranged from 4.84 to 8.83 m$^2$ ha$^{-1}$, a much narrower range (mean 6.32 ± 0.23 m$^2$ ha$^{-1}$, CV 17.1%, $n = 24$) and a more uniform stand growth.

Overall, evidence is clear that early post-planting management regime which promotes early growth, vigor and survival during the first 2 years after planting is a prerequisite for achieving high production at harvest and improved P nutrition promotes uniform stand growth.

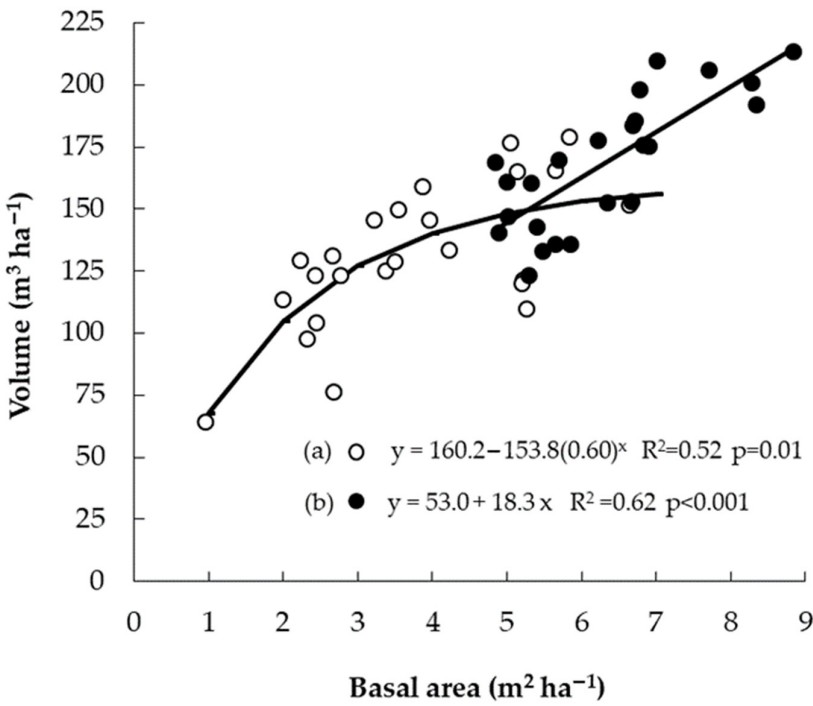

**Figure 9.** *Eucalyptus pellita*: relationship between basal area at age 2 years and volume at age 6 years; (**a**) without P (open circle), (**b**) with 12 kg P ha$^{-1}$ (closed circle). Data points represent individual plots from all replications and respective treatments from six sites.

## 4. Discussion

### 4.1. Productivity of Successive Rotations

In Sumatra, Indonesia, *A. mangium* dominated the plantation forestry landscapes and fiber supply for the pulp and paper industry until about 2012. Research demonstrated that with the application of integrated management practices, growth rates in the second rotation were in the range of 40.0–49.0 m$^3$ ha$^{-1}$y$^{-1}$, substantially higher than the first rotation [2]. There have not been any major soil nutrient deficiencies, apart from the need to apply a small dose of P (about 10 kg ha$^{-1}$) at planting [15,26]. Furthermore, if the weeds are managed with herbicide initially, stands grew fast enough with large crowns to close canopy by age 1.5–2.0 years and this process suppressed the weeds including *Imperata* grass. However, operational outcomes did not reflect the experimental results showing productivity increase from the first to the second rotation. For example, estate wide pre-harvest, inventory of the second rotation stands showed yields similar to the first rotation, but no consistent increase [1]; this was partly because of the constraints in the quality and timeliness of operations and applicability of some of the research results [7]. Initially, when evidence of yield decline appeared in parts of the estate, company managers were reluctant to accept that situation, and shortfalls in wood supply were partly met by shortening the rotation and over-cutting [E.K.S Nambiar, personal discussions with managers and observations]. The escalating threats of diseases in plantations forced managers to change the species from *A. mangium* to *E. pellita*, at a rate rarely seen in plantation forestry. In a span of

few years, the landscapes carrying plantation forestry in Sumatra changed exclusively from acacia to eucalypts.

This history and the accompanying changes in ecosystem stresses and processes have impacted on the growth rates across successive rotations managed over nearly 30 years (Figure 1) and this has major impacts on wood supply [7]. It should be noted that the MAIs (Figure 1) reported for the four rotations are not based on growth measured at a common age, especially R-3 which was terminated earlier. However, it is worth noting that at age 3 years, the MAIs of R-2 and R-3 were close, 29.3 and 27.6 $m^3$ $ha^{-1}$ $y^{-1}$, respectively, indicating the capacity of the local soil to support long term productivity of successive rotations, and that pests and diseases, rather than changes in soil properties, are the main threats to sustainability in this case. Comparison of growth rates of R-2 (*A. mangium*) and R-4 (*E. pellita*) at the same site show that the growth rate of *E. pellita* is significantly lower than that of *A. mangium*. Thus, Figure 1 displays the rise in growth rates from the first to the second rotation, the fall due to diseases in the third rotation and the path to recovery in the fourth rotation by changing species (a management option). It illustrates the consequence of management responses to ecosystem threats and processes over the long term. While these results are from a long-term experiment, our experience shows that this trend occurred in large areas in south and central regions of Sumatra. A somewhat similar pattern was described in the productivity of three successive rotations of *Pinus radiata* in 30+ year cycles in south Australia, characterized by a conspicuous "2-R decline" and a strong recovery in the third rotation, all together over a century. The decline in the pine ecosystem was largely attributed to unsustainable management practices, especially the loss of organic matter. The decline was reversed and productivity increased in broad scale forestry by the outcomes from coherent multi-disciplinary R&D and integrated application of best management practices operationally [25]. In the humid-tropical-equatorial environment with fast growing short rotations, such dynamics in productivity can occur in shorter time frames (Figure 1) [11].

Tropical short-rotation plantation forestry has an advantage compared to temperate forestry which are managed over long cycles: it can change and deploy new germplasm at short intervals as a tool to manage or avoid ecosystem threats, improve productivity or to widen product options to suit the markets. Such changes can be at the species level, within species or as intra-species hybrids. It is important that such changes are based on the outcomes from systematic genetics and breeding programs well grounded in the local environment (e.g., *E. pellita* in Sumatra and Kalimantan), not by deploying imported materials without rigorous local testing. However, both deployment of new germplasm and implementation of sustainable management should go hand in hand for successful forestry.

The removal or retention of slash and litter at three inter-rotation phases (legacy effects) had statistically significant, but not large effects on the growth of *E. pellita*, but retention of organic matter and additional soil P supply with fertilizer application (Figure 2) accelerated the growth improvement consistently from age 1 year to harvest (Figure 3). These results are closely similar to the results with *A. auriculiformis* in South Vietnam where conservation of site organic matter and application of P in the planting hole, as in this study, increased productivity substantially at age 7 years [12]. At the start of the second rotation, the slash and litter retained from the previous rotation contained 13.7 kg P $ha^{-1}$ [2]. Further recycling of P, mediated by mineralization and uptake from the litter and soil, would have occurred during the following rotations. The consistently higher concentration of extractable P in the surface soil under high organic matter retention ($BL_3$) compared to that under $BL_0$ treatment (Figure 4), suggests that there has been a transfer of P from surface organic matter to the soil, but we note that nutrients are also transferred in the soil through fine root turnover and decomposition. Extractable P has been found to be declining during a rotation of acacia [2–21]. Slash and litter retention in eucalypt plantations in Brazil affected the various pools of P in the soil [27]. While we do not yet have a good understanding of the dynamics of soil P and its relationships with stand growth in

our experimental ecosystem, it is important to recognize from a management perspective, that even when site organic matter was conserved, incorporation of fertilizer P in the planting hole and increasing the supply in the immediate vicinity of the roots improved wood volume at harvest. It is likely that at this site and in Vietnam [13], the response to P may not have occurred if the same amounts of P were broadcast over the slash-litter layer. In soils with low pH (typically pH 4.1–4.4, Table 1) and high P fixing properties, an increase in the nutrient flux density at the root–soil interface may have been a key process for promoting high uptake and growth.

In plots from where slash and litter was removed repeatedly (BL$_0$), the growth rates were 46.6 m$^3$ ha$^{-1}$ y$^{-1}$ in *A. mangium* and 29.4 m$^3$ ha$^{-1}$ y$^{-1}$ in *E. pellita* (Figures 2 and 3). These relatively high growth rates demonstrate the productive capacity of some sites and soils in Sumatra to support good growth of plantations [2,15,28]. In contrast, synthesis of results on eucalypts from several comparable studies in the tropics and subtropics show significant loss of yield when organic matter was removed during the inter-rotation management from sites [11,12,29]. Such adverse effects tend to be highly pronounced in light textured, low nutrient soils [30], and the effects can be nutrient specific as illustrated by the pronounced effects on the supply of and response to K [27,29,31].

We used the two extreme management treatments (repeated organic matter removal-BL$_0$ and doubling slash addition (BL$_3$) to examine the long term impacts on soil properties because earlier measurements showed little difference between other treatments. High organic matter retention appeared to reduce soil pH slightly, notably at the end of R-3 when slash load was abnormally high because of low recovery of merchantable wood and dead trees. In practice, there is little or no chance for the occurrence of BL$_3$ level slash load at successive rotations. The planting of eucalypts seems to have reversed the general trend of declining pH with acacia rotations, as seen by the rise in pH by 0.12–0.25 units after one rotation of eucalypts. This warrants further investigation including more comprehensive understanding of nutrient cycling. Results from experiments [11,30] as well as review [12] show that changes in soil pH, while tending to fluctuate seasonally and over the years, are unlikely to be an ecosystem risk in short rotation forestry.

The changes in soil organic carbon (SOC) and total N are nearly parallel; it appears that small increases in them are possible (by 10–12%) over the long term, if management followed conservation of site resources and zero tillage (Figure 4). There was no evidence of a decline in SOC and N as a result of slash and litter removal (soil in our study site had 58% clay), similar to the results in *E. globulus* at a fertile site in Western Australia [32]. In contrast, on a poor-sandy soil in Congo, after two rotations of *Eucalyptus* plantation, the complete removal of slash and litter reduced the SOC in the top soil (0–5 cm) by 44% with marked reduction in yield [33]. The N fixing capacity of *A. mangium* plantation in this area was estimated (using N$^{15}$ techniques) to vary between 26 and 142 kg N ha$^{-1}$ at age 18 months, depending on factors including soil, P supply and tree growth rates [34]. None of the experiments with first and second rotation *E. pellita* have so far shown a response to N application in this region [15,35], but this may change with time. Conservation of site resources and minimum tillage has been a common practice in managing eucalypts plantations in Brazil for many years [36]. Evaluation of a large number of commercial plantation sites in Brazil revealed that over consecutive rotations, SOC changed little in the 0–15 cm layer and decreased slightly in the 15–30 cm layer [37].

Caution is necessary in extrapolating the results in Figure 4, broadly, because these results are confined to one productive site (Figure 1). In South Sumatra, even among the sites used in this research, soils varied considerably in critical properties (Table 1), for example, depth of A-horizon ranged from 3 to 15 cm, depth to plinthite (hence the rooting volume) from 34.2 to 109.9 cm, in addition to the large differences in soil textural components, nutrient status and tree growth rates (Figure 5). The critical importance of conserving site organic matter and nutrients is now well recognized as key for sustainable wood production and maintenance of ecosystem processes in short rotation forestry [12,13,30]. In Sumatra, retention of slash and litter and minimum tillage are by far the most common

inter-rotation practice implemented by the forestry industry, as well as the application of P fertilizers at each planting cycle. Systematic and regular examination of the operational inventory is essential to detect changes in productivity and its spatial patterns promptly in order to foresee problems and revise management accordingly [7].

### 4.2. Managing Productivity across Diverse Sites

In this experiment on a productive site, in the first rotation of *E. pellita* the lowest and the highest MAIs were 22.5 and 34.9 $m^3$ $ha^{-1}$ $y^{-1}$, compared to 43.1 and 51.7$m^3$ $ha^{-1}$ $y^{-1}$, respectively, achieved by *A. mangium* in the second rotation (Figure 1; [2]), so the overall wood volume productivity of eucalypts is on average 43.3% lower (Figures 1 and 3). A similar pattern was seen in operational pre-harvest inventory in the company estate: the growth rates of *E. pellita* on sites which previously carried typically two rotations of *A. mangium* were in the range 15.6–17.6 $m^3$ $ha^{-1}$ $y^{-1}$, significantly lower than that of *A. mangium* which were in the range of 22–35 $m^3$ $ha^{-1}$ $y^{-1}$ [1]. In a more recent inventory of the current plantations in the same area (unpublished MHP data), using 283 permanent sample plots, MAIs ranged widely from 15.1 to 39.8 $m^3$ $ha^{-1}$ $y^{-1}$ (mean, 25.2 $m^3$ $ha^{-1}$ $y^{-1}$). Distribution of plots in productivity classes were 25% in 15–20 $m^3$ $ha^{-1}$ $y^{-1}$, 24% in 20–25 $m^3$ $ha^{-1}$ $y^{-1}$, 29% in 25–30 $m^3$ $ha^{-1}$ $y^{-1}$, 16% in 30–35 $m^3$ $ha^{-1}$ $y^{-1}$, and 7% more than 35 $m^3$ $ha^{-1}$ $y^{-1}$. These results suggest that there have been improvements in productivity since the results from the first inventory [1]. As mentioned in the introduction, overall, the assessments within the company estate show that the standing volumes of eucalypts are 24–36% lower than in those obtained in the best rotations of *A. mangium* across sites. Thus, the overall productivity of eucalypts poses a challenge to wood access for local processing and the regional economy, and therefore, the need for improving productivity sustainably is a high priority management goal for the regional industries. In the following sections we explore the options for improving the productivity in South Sumatra.

### 4.3. Management of Phosphorus Nutrition

The strong response to P even when slash and litter were retained repeatedly (Figure 2), and P responses in other studies up to age 3–4 years [15,26] prompted the need to test and describe the growth responses to P at representative sites in the region. The Bray-1 extractable P in soils is generally low (Table 1) and tends to decline after the first rotation (Figure 4), but so far there has been no work validating the relevance of this or other extractable pools of P as diagnostic tools to assist the decision making process on P application to forest plantations in Indonesia. Studies in Sumatra [2,15] and in other ecosystems [27] had shown that early response to P, often vigorous in the first year or two, dissipates steadily with stand age and thus the relative response declines from age 1 to 6 or 7 years of age. Therefore, we wanted to define the response over the full rotation period, so that the results are closer to operational outcomes and will be more convincing to managers for adoption. As with other studies, here also relative responses declined with age (data not shown) at all sites: for example, at a high productivity site the P response decreased from 71.4% at age 2 years to 44.4% at age 6, and at a low productivity site the corresponding drop was from 123.7% to 41.1%, although the positive responses to P remained significant at the end of rotation and the relative response depended on site (Figure 5).

A number of processes can contribute to this decline in relative response with stand age. Application of P close to the fast developing root system can stimulate root growth, and as the stand develops the soil volume accessible for P uptake, away from the localized P fertilized zone, will increase and thus trees may have more access to soil P. The development of root system and associated mycorrhizae may also improve access to P pools, previously considered "non-labile" [31,38]. Secondly, plantation tree species including eucalypts can meet a significant share of P requirement for new growth from internal re-translocation especially if the nutrient contents in live tissues are high, from green leaves before and during senescence [39,40]. In a study with *Eucalyptus* on a soil low in fertility in Congo, nutrient requirements for early growth were supplied from soil, but after the

canopy closure about 30% of P demand was met by re-translocation [41]. At the *E. pellita* site described in Figure 2, we found that nutrients re-translocated from senescent leaves as a proportion of the initial amounts amounted to P (63.9%), N (29.0%), K (44.6%) and Ca (none) [42] consistent with the results reported in *E. globulus* in South Australia [40]. The current uptake of nutrients and internal re-translocation are processes that occur concurrently, not sequentially, and nutrients derived from both sources are used for new growth [39]. These processes influence the direct response to added nutrients.

This regional network study represents ex-acacia sites where slash and litter were retained for at least two rotations and P was routinely added at planting, productivity ranged from 24.4 to 35.6 m³ ha⁻¹ y⁻¹. The clear conclusion is that a single application of P (incorporated in the planting pit) gave additional wood yield, ranging from 16 to 66% over the control, at harvest. At high productivity sites, where growth rates in P-0 were within 22–26 m³ ha⁻¹ y⁻¹, response to P maximized at an application rates ranging between 24.0 and 25.4 kg ha⁻¹; in contrast at low productivity sites, where MAIs were in the range of 15–21 m³ ha⁻¹ y⁻¹, P response maximized at a dose of 6.8–7.8 kg ha⁻¹ (Figure 5). These findings have considerable relevance to forest operations because additional wood yield by application of P ranged from 20.5 to 80.6 m³ ha⁻¹, the mean across all sites being 41.4 m³ ha⁻¹, an important amount when scaled up to annual wood flow to the mill. The results (Figure 5) also provide a basis for refining the current operational practices for managing P fertilization matched to sites.

### 4.4. Site and Wood Production

The development of a hard plinthite horizon in the profile is part of soil genesis of Ultiosols and is a common feature in some soils in South Sumatra. The growth of *E. pellita* was related to the depth to plinthite horizon (Figure 7a), the deeper this layer the higher the growth; between the extremes of depth there was nearly a 3-fold difference in wood volume. A similar relationship between the subsoil morphology including plinthite and growth has been found in *A. mangium* in these landscapes [22]; that study also observed very low fine root activity in the plinthite horizon. In *A. mangium* at the site where plinthite layer was at 40 cm, MAI of was 18.0 m³ ha⁻¹ y⁻¹, whereas at site where plinthite layer was below 70 cm MAI was 44.0 m³ ha⁻¹ y⁻¹ [43]. These types of soil horizons are commonly high in Fe and exchangeable Al, P-fixing capacity and are known to affect crop productivity [44].

The relationship between the depth of the A-horizon and production was not strong, although there is a trend (Figure 7b). This suggests that the depth to plinthite which has a strong influence on total rooting volume, and thus the availability of stored water in the low rainfall-dry months, may well be a critical factor determining growth over the rotation [35]. As noted in results, survival was not influenced by the depth of A-horizon or depth to plinthite layer, indicating that the soil depth (rooting volume) influenced production primarily via growth rates. Nutritional deficiencies in the top soil layer can be partly or fully corrected by fertilizer additions, as shown with P (Figure 5), but constraints imposed by the soil profile features such as plinthite are not easily manageable. Techniques such as deep ripping to the plinthite to allow cleavages for root growth would be a drastic, expensive and highly soil disturbing operation involving heavy equipment. A study in progress is mapping spatial distribution of productivity and key soil parameters to guide the organization's investment decisions in deploying improved genetic materials and appropriate management operationally, and to set R&D priorities.

### 4.5. Stand Attributes and Production

Ongoing tree mortality with age is an endemic problem in the tropical plantation environment in SE Asia and final stocking is a significant yield determinant in short rotation forestry [1]. In the company's operational forestry, at age 6 years, survival ranged between 55.6% and 93.1% with a mean of 72.3% [45], suggesting high variability across sites, as were the cases in this study (Figure 6). The high mortality in the first year after

planting (Figure 6) suggests possible water stress in the dry months as a reason for seedling death, because our field visits did not indicate any pest and disease infestation at any site. Windthrow can cause damage in this ecosystem as the trees grow tall, especially if the tree crowns are heavy.

The strong relationship between stocking (number of trees ha$^{-1}$) at harvest and volume, especially when trees have adequate P supply (Figure 7a) highlights the need to ensure good survival in order to receive the best returns from P application. A comparison of the slopes of the two regression lines with and without P, show how at a given final stocking, adequate P supplies have improved volume gains in trees. There were no discernable links between the soil attributes in Table 1 and survival. However, this project was not set up to follow the post planting stresses and the mechanisms and processes causing the loss of seedlings and young trees.

One way to compensate for the tree losses during the rotation is to plant at a higher stocking rate, and that option is currently under investigation. Genetic selection weighted more towards survival and resistance to windthrow are possibilities. Overall, the results highlight the importance of new research to understand the site specific causes of mortality and developing management regimes to maximize survival [46].

The early growth of basal area determined by the stem diameter and stocking (trees ha$^{-1}$) was a reliable measure of the stand establishment, although at some of the sites in the region mortality tends to continue beyond age 2 years (Figure 6). The importance of this for final harvest volume is well illustrated in Figure 9. The relationship is clearly segregated between P-deficient and P-sufficient stands, representing the diminishing CAI with age under P deficiency and a progressive linear growth with P sufficiency. As described in the results, the differences in the range of basal area (Figure 9) between P deficient and sufficient plots leads to the conclusion that adequate P nutrition reduced variations in productivity across the area, creating a more uniform stand which is an important management objective. A similar relationship between early basal area and final volumes were found in A. *auriculiformis* and Acacia hybrid in contrasting sites in South and Central Vietnam [13]. There may be opportunities for further developing these relationships (Figures 7–9) into predictive models for enabling site specific management decisions and appropriate interventions. Research in this region showed that application of second dose of fertilizers beyond planting at "mid-rotation" stage have not improved productivity [16], suggesting that in short rotation forestry, remedial actions to correct omissions or failures in management may not be practical. We conclude that in this environment, application of the best possible management to achieve the best growth rates and survival early in the rotation (establishment phase) is central for achieving high production.

### 4.6. Prospects for Increasing Productivity

In the introduction, we identified the current wood deficit for the pulp and paper mills in Sumatra. Since industry is unlikely to expand the plantation area, it is essential that productivity per unit area in their current estate is increased, sustainably, for managing the fluctuations in production that are inevitable in response to vagaries of climate and stresses, and to ensure a stable wood supply [7]. To put this challenge in perspective, some points are worth summarizing here. While the current level of productivity of *E. pellita* is variable and in general lower than the best rotations of *A. mangium*, the research results presented here and discussed elsewhere [7,15,16] show several management options and opportunities for increasing productivity of *E. pellita*.

Based on detailed analysis of the constraints and opportunities [7,47], actions which would provide prompt returns include (i) application of integrated package of practices derived from the current knowledge [11,12,20], (ii) investments in adaptive research to support operations, (iii) critical analysis of the pre-harvest inventory to understand the spatial and temporal variations in growth rates and their reasons and (iv) importantly adequate and systematic attention to improve the working conditions, skills and motivation of labor force [7].

Another pathway for increasing wood supply is to recruit, retain and successfully manage a large network of small growers as partners. The Government of Indonesia (GoI) has had an active program since 2015, with a target of distributing 12.7 million ha of land to thousands of farmers and smallholders in the forest-rural landscape. Those who receive the land have the legal rights to it and use the land for the production of food and various commodity crops including wood. As of 2019, the program has allocated 1.21 M ha in Sumatra and 1.32 M ha in Kalimantan [48]. The goal of the forestry part of this program is to increase national wood supply for a variety of end uses, strengthen rural livelihood and to avoid further deforestation. Already about 25,000 small growers in Sumatra and about 4400 small growers in Kalimantan (estimated from diverse sources) have joined forestry-partnerships with companies for growing *E. pellita,* indicating community interests. Indonesia has a long tradition of small grower forestry, especially in Java [49] and our field experience and a recent study [50] confirm that there are significant areas with soils and site characteristics suitable for *E. pellita*. There are also options for agroforestry with commercial tree species and food crops [51]. Commercial tree growing in farm holdings of 1–5 ha are profitable for resource poor rural communities in SE Asia [52,53] and such developments along with local value adding is a path to rural development and poverty alleviation in the tropics [54]. An analysis of the steps required to strengthen the engagement of small growers in forestry is not within the scope of this paper, but the constraints have been identified by other studies [55,56]. However, we make few comments. An earlier assessment has found that the growth rates of *A. mangium* among the company "out grower" program participants were only 50% of the rates achieved in the company estate [43]. This is not an acceptable outcome for anyone because fast growth rates, high productivity, log quality and early returns are incentives for the active engagement of local communities. Although the research reported in this paper and many others were carried out in the company land with their support for practical and logistical reasons (e.g., long term access to and tenure of experimental sites, operational and technical help), the results are freely available to all growers. The sites and soils in this research are within the catchment of many small growers. Additional steps required for gaining greater engagement of small growers may include setting up on farm-demonstrations with growers' inputs, which can illustrate the value of the operationally feasible best bet-package of practices, better productivity and value [53]. This needs to be supported by fair price, reliable access to wood market, and modest investment support.

## 5. Conclusions

The productivity of four successive rotations (three *A. mangium* and one *E. pellita*), measured over 30 years, demonstrate the dynamic nature of productivity: increased production of *A. mangium* from first to the second rotation due to improved management, loss in production caused by diseases in the third rotation and growth recovery following a change of species to *E. pellita*. This illustrates the importance of long-term research covering full rotations in the field for understanding sustainability. For managing the current wood deficits and setting the course to sustainable production for the current estate of *E. pellita*, a number of factors and their contributions to productivity should be understood. Judicious inter-rotation management including conservation of organic matter combined with application of P fertilizer are important inputs to achieve high productivity. Long term changes in soils in the 0–10 cm soil show relatively small changes in pH, organic carbon, total N and extractable P. Application of additional P at planting yielded higher wood volumes at harvest at all sites, despite the history of organic matter conservation and P fertilizer applications to all previous rotations. The amount of P to be supplied to obtain 95% of the maximum yield was strongly dependent on the overall productivity level of sites, with higher productivity sites responding to greater P application rates. P deficiency induced high variability in stand growth and improved P nutrition through a single application of an adequate amount of P-fertilizer at planting also promoted much greater uniformity in the stand. The depth of occurrence of the plinthite layer in the profile

impacts on productivity strongly; the deeper the layer the higher the growth rates and vice versa. Rapid early growth and maximum survival lead to a good basal area by age 2 years and this is a guide to production at harvest. Tree survival is generally low and a critical challenge in this environment and therefore process-based research to understand and manage survival is a priority. Based on current knowledge, several management options are available which can be adopted to increase the productivity of *E. pellita* broadly in the next rotations, both in the company estate and in the small holdings owned by farmers and others for managing the future wood supply in Sumatra.

**Author Contributions:** Conceptualization, E.B.H., M.A.I. and E.K.S.N.; methodology, E.B.H., M.A.I. and E.K.S.N.; validation, E.B.H., M.A.I. and E.K.S.N.; formal analysis E.B.H., M.A.I.; investigation, E.B.H., M.A.I., E.K.S.N.; data curation, M.A.I. and E.B.H.; writing—original draft preparation, E.B.H.; writing—review and editing E.K.S.N., E.B.H.; project administration, M.A.I. and E.B.H.; funding acquisition, M.A.I. All authors have read and agreed to the published version of the manuscript.

**Funding:** This research was financially supported by PT. Musi Hutan Persada, South Sumatra.

**Data Availability Statement:** Data is unavailable.

**Acknowledgments:** We are grateful to the management of PT. Musi Hutan Persada, South Sumatra for their commitment to support long term research and publication of results. We thank R&D Manager and Silviculture Team at the Research and Development, PT. Musi Hutan Persada for various support, assistance in experiment works, long-term maintenance and data collection. Daniel Mendham (CSIRO) and Philip Smethurst (CSIRO) reviewed the earlier draft and suggested several improvements. The first two authors thank Sadanandan Nambiar (CSIRO) for his long-term support.

**Conflicts of Interest:** The authors declare no conflict of interest.

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
