# Peer review of "Productivity of Eucalyptus pellita in Sumatra: Acacia mangium Legacy, Response to Phosphorus, and Site Variables for Guiding Management"

_forests, doi:10.3390/f12091186_

Round 1
Reviewer 1 Report
Review of “Productivity of Eucalyptus pellita in Sumatra: Acacia mangium legacy, response to phosphorus, and site variables for guiding management”.
The paper studied the productivity of Acacia mangium and Eucalyptus pellita plantation in Sumatra across 30 years including three rotations of Acacia mangium followed by Eucalyptus pellita. Different aspects of forest management plantation are approached. First the authors examine the effects of inter-rotation slash and litter management applied to acacia on E. pellita growth and secondly, evaluate the impact of additional phosphorus and the plinthite layer depth on E. pellita productivity across 6 sites. The objective of the study is to give different ways to increase the productivity per unit area across a package of best practices.
I think it is a well written and complete article which brought lot of responses for Sumatra tropical forest management.
I have only one minor concern about the form of the discussion part and the different sections presented.
I do not see a real difference between the title of the topic 4.4 and 4.5.
L 603 : 4.4. Site and Stand Attributes and Wood Production
L 629 : 4.5. Stand Attributes and Production
Each section deals with different subjects but the title of each one is quite similar. The first one (4.4) is more about the impact of the plinthite layer and soil horizon A depth on E. pellita productivity and the second one on the survival rate.
I recommend to the authors to be more specific.
Author Response
Response to Reviewer # 1
Comments and Suggestions for Authors
Review of “Productivity of Eucalyptus pellita in Sumatra: Acacia mangium legacy, response to phosphorus, and site variables for guiding management”.
The paper studied the productivity of Acacia mangium and Eucalyptus pellita plantation in Sumatra across 30 years including three rotations of Acacia mangium followed by Eucalyptus pellita. Different aspects of forest management plantation are approached. First the authors examine the effects of inter-rotation slash and litter management applied to acacia on E. pellita growth and secondly, evaluate the impact of additional phosphorus and the plinthite layer depth on E. pellita productivity across 6 sites. The objective of the study is to give different ways to increase the productivity per unit area across a package of best practices.
I think it is a well written and complete article which brought lot of responses for Sumatra tropical forest management.
Response: Thank you
I have only one minor concern about the form of the discussion part and the different sections presented.
I do not see a real difference between the title of the topic 4.4 and 4.5.
L 603 : 4.4. Site and Stand Attributes and Wood Production
L 629 : 4.5. Stand Attributes and Production
Each section deals with different subjects but the title of each one is quite similar. The first one (4.4) is more about the impact of the plinthite layer and soil horizon A depth on E. pellita productivity and the second one on the survival rate.
I recommend to the authors to be more specific.
Response: Thank you. We accept the suggestion and have revised them accordingly. Sub-heading 4.4 has been revised to become 4.4. Site Attributes and Wood Production while Sub-heading 4.5 has been revised to become 4.5. Stand Attributes and Wood Production
Reviewer 2 Report
* title is misleading as it focuses on E. pellita, while the paper is mostly about A. mangium
* productivity should be assessed for the same age, especially when you consider fast-growing species; cant compere 10-years-old A. mangium and 6-y-o E. pellita!
M&M section has to be updated on the info about comparisons of productivity of 6-y-o species
* soil conditions - you bulked samples from 0-10 cm layer to describe soil conditions, at Niru site A horizon is 3 cm only, so you catch two horizons in 0-10 layer, hence the soil characteristics is wrong
* why you use ANOVA - have you checked the normality?
* Unnecessary repetition of results in discussion
Some parts of the discussion are slightly related to the main subject of the paper, those should be omitted (e.g. lines 689-721)
* I am not convinced to citing unpublished data - if you have it, you shoud present it, otherwise it is a bit blurry IMO
Author Response
Response to Reviewer # 2
Comments and Suggestions for Authors
* title is misleading as it focuses on E. pellita, while the paper is mostly about A. mangium
Response: We are surprised about this comment, and we do not agree with the reviewer’s statement. The title of the paper represents the objectives and experimental work done. It explains that the productivity of Eucalyptus pellita is the focus of the paper. In the introduction, we described quite clearly about A. mangium management to provide an adequate background of the development and productivity of A. mangium plantation in Sumatra before the species was replaced with Eucalyptus pellita, because planting of A. mangium was no longer economically viable due to widespread wilt disease caused by Ceratoystis fungi. We have presented productivity trends over the long term, and the sustainability challenges. The legacy effects of A.mangium on E.pellita is a key aspect of the plantation ecosystems in Sumatra
In our study, one experiment (successive four rotations) was planted with E. pellita on the same site, where in the previous three rotations different levels of slash and litter management of A. mangium were applied. The legacy effects of previous rotation A. mangium on the productivity of E. pellita were assessed. All other six experiments were about E. pellita, on different sites previously grown with 2-3 rotations of A. mangium. The productivity of A. mangium of the previous rotations was reported for the purpose of comparison with E. pellita and the long term trend. In addition, the legacy effects of A. mangium on the productivity of E. pellita and changes of soil properties were presented and discussed. These are the realities for much of plantation estate in Sumatra. We point out that information on management of E.pellita is the major part of this paper (e.g. see Figures 2, 3, 5, 6, 7 and Table 2).
* productivity should be assessed for the same age, especially when you consider fast-growing species; cant compere 10-years-old A. mangium and 6-y-o E. pellita!
M&M section has to be updated on the info about comparisons of productivity of 6-y-o species
Response: Yes, we did compare the productivity of A. mangium and E. pellita at a common age, namely at age 6 years to have a fair comparison (see Figure 1; Table 2). One purpose of using growth rates (MAI) is to allow comparison in cases where there are differences in stand age; this is a common approach in research. We consider that it is unnecessary to describe this in the materials and methods section.
* soil conditions - you bulked samples from 0-10 cm layer to describe soil conditions, at Niru site A horizon is 3 cm only, so you catch two horizons in 0-10 layer, hence the soil characteristics is wrong
Response: Soil samples were taken from every plot and replication at six sites at 0-10 cm of soil depth consistently without considering the soil horizon to provide good and fair comparison of soil properties between sites. Niru had a shallow A-horizon (3 cm depth) which was obviously one of soil physical characteristics of the site. There is nothing “wrong” with that.
* why you use ANOVA - have you checked the normality?
Response: ANOVA can be used when we want to compare differences between the treatment means of an experiment having more than two treatments. It is a valid statistical procedure for experimental design we used . All our experiments had five levels of treatment, applied in randomized replications; it is certainly valid to apply ANOVA. Data did not violate the normality required by ANOVA. We have carefully checked the data for normality and outliers prior to analyses. These are standard procedures.
* Unnecessary repetition of results in discussion
Some parts of the discussion are slightly related to the main subject of the paper, those should be omitted (e.g. lines 689-721)
Response: It is not clear what the reviewer means with the comment “unnecessary repetition of results in discussion”, without identifying which parts of the discussion are repetition of the results. In the result section we presented what we observed in the experiment, while in the discussion section we tried to explain the bio-physical bases of the findings as well as practical implications for future management of the plantation. It is nearly impossible to discuss results, without reminding the reader some leads into the results.
We do not think it is necessary to delete part of the discussion (line 689-721). These sentences are to provide the prospect for increasing productivity of E. pellita, which is still low in the region, and we need to explore the management options for improving production, including for small-growers. Wood deficit is a major issue for industry in Sumatra.
* I am not convinced to citing unpublished data - if you have it, you should present it, otherwise it is a bit blurry IMO
Response: We did present data when we cited unpublished report in the text, for example data of the recent inventory of permanent sample plots at MHP company. It is not practical to publish company inventory; our aim is to point out the opportunities. Sometimes, it is not feasible to cite only from published paper or report. In the paper we wanted to provide the best and relevant information available even though the data was from an unpublished (but based on our experience, highly reliable) source. We have used these “unpublished” information, not to substantiate our results, but to point out the direction for future from a management and application perspectives.